# Functionally and Metabolically Divergent Melanoma-Associated Macrophages Originate from Common Bone-Marrow Precursors

**DOI:** 10.3390/cancers15133330

**Published:** 2023-06-24

**Authors:** Gabriela A. Pizzurro, Kate Bridges, Xiaodong Jiang, Aurobind Vidyarthi, Kathryn Miller-Jensen, Oscar R. Colegio

**Affiliations:** 1Department of Biomedical Engineering, School of Engineering and Applied Science, Yale University, New Haven, CT 06511, USA; kate.bridges@yale.edu (K.B.); kathryn.miller-jensen@yale.edu (K.M.-J.); 2Department of Immunobiology, School of Medicine, Yale University, New Haven, CT 06511, USAaurobind.vidyarthi@yale.edu (A.V.); 3Department of Molecular, Cellular and Developmental Biology, Yale University, New Haven, CT 06511, USA; 4Department of Dermatology, School of Medicine, Yale University, New Haven, CT 06511, USA; oscar.colegio@roswellpark.org; 5Department of Dermatology, Roswell Park Cancer Comprehensive Center, Buffalo, NY 14203, USA

**Keywords:** tumor-associated macrophages, melanoma, tumor microenvironment, macrophage ontogeny, immunosuppression, tissue-resident macrophages, metabolic pathways

## Abstract

**Simple Summary:**

Melanoma, one of the deadliest cancers, presents challenges due to the incomplete understanding of the key mechanisms driving its aggressive behavior. High numbers of macrophages in the melanoma microenvironment are associated with poor outcomes. Macrophages are immune cells that act as a double-edged sword, supporting tumor growth but holding the potential to re-activate the anti-tumor response. Previous studies on other cancer models have linked the tumor-supporting roles of tumor-associated macrophages with their ontogeny. In this report, we investigated macrophage infiltration in a melanoma model to understand their origin, evolution, activation profile, and association with the immunosuppressive microenvironment. Unlike other models, we found a common origin in the two main macrophage subsets, which varied over time and differed in their functional and metabolic profiles. These findings shed light on the features and evolution of melanoma-supportive macrophage subsets, which could be used to discover new ways of targeting these cells therapeutically.

**Abstract:**

Tumor-associated macrophages (TAMs) can be widely heterogeneous, based on their ontogeny and function, and driven by the tissue-specific niche. TAMs are highly abundant in the melanoma tumor microenvironment (TME), usually correlating with worse prognoses. However, the understanding of their diversity may be harnessed for therapeutic purposes. Here, we used the clinically relevant YUMM1.7 model to study melanoma TAM origin and dynamics during tumor progression. In i.d. YUMM1.7 tumors, we identified distinct TAM subsets based on F4/80 expression, with the F4/80^high^ fraction increasing over time and displaying a tissue-resident-like phenotype. While skin-resident macrophages showed mixed ontogeny, F4/80^+^ TAM subsets in the melanoma TME originated almost exclusively from bone-marrow precursors. A multiparametric analysis of the macrophage phenotype showed a temporal divergence of the F4/80^+^ TAM subpopulations, which also differed from the skin-resident subsets and their monocytic precursors. Overall, the F4/80^+^ TAMs displayed co-expressions of M1- and M2-like canonical markers, while RNA sequencing showed differential immunosuppressive and metabolic profiles. Gene-set enrichment analysis (GSEA) revealed F4/80^high^ TAMs to rely on oxidative phosphorylation, with increased proliferation and protein secretion, while F4/80^low^ cells had high pro-inflammatory and intracellular signaling pathways, with lipid and polyamine metabolism. Overall, we provide an in-depth characterization of and compelling evidence for the BM-dependency of melanoma TAMs. Interestingly, the transcriptomic analysis of these BM-derived TAMs matched macrophage subsets with mixed ontogeny, which have been observed in other tumor models. Our findings may serve as a guide for identifying potential ways of targeting specific immunosuppressive TAMs in melanoma.

## 1. Introduction

Melanoma is the most lethal form of skin cancer. Approximately 200,000 new cases of melanoma will be diagnosed in 2023 in the US, with more than half being diagnosed at an invasive stage [1]. Tumor control is particularly challenging for a host, since it is a sterile insult yet broadly pathogenic. Several components of the tumor microenvironment (TME) are closely related to cancer progression through multiple mechanisms facilitating dissemination and contributing to immune suppression [2].Despite being an aggressive disease, melanoma is an immunogenic tumor; therefore, immunotherapy, which aims to reprogram the immunosuppressive TME and boost T cell function, has provided a promising approach for melanoma treatment [3,4]. 

Macrophages constitute the dominant myeloid cell population in most solid tumors and studies on both mice and humans have linked macrophage density to tumor growth [5]. Tumor-associated macrophages (TAMs) can facilitate tumor progression, proliferation, and metastasis by stimulating angiogenesis and inhibiting antitumor T cell responses. Previous results have also shown that immunosuppressive TAMs can be functionally reprogrammed to help control tumor growth [4,6]. TAMs were historically considered to be in an “M2 state”, referring to the dichotomy of pro-inflammatory (M1) vs. anti-inflammatory (M2) cells [5]. Although this classification works well in vitro, macrophages within a tissue microenvironment acquire a wide spectrum of activation states influenced by multiple factors and environmental cues [7]. In an in vitro 3D collagen model, we previously showed that stromal and tumor cells shape the macrophage activity in the early melanoma TME. These cell–cell interactions induced bone-marrow (BM)-derived macrophages to acquire an immunosuppressive functional signature that resembled TAMs from melanoma tumors, which differed from those canonical polarization states [8].

There have been efforts to understand the sources of heterogeneity in TAM behavior and how to harness them therapeutically. In recent years, the classification of mononuclear phagocytes has been revisited, particularly macrophage ontogeny [9]. The expression levels of F4/80 and CD11b during development identified BM-dependent and -independent tissue-colonizing macrophages [10]. Tissue-resident macrophages have been shown to have a distinct origin, gene expression, and phenotype from those derived from CC-chemokine receptor 2 (CCR2)^+^ monocyte precursors recruited to a site of inflammation. The new mononuclear phagocyte classification system, based on cellular origin, allows a robust definition across tissues and species [9] and a better understanding of their functional roles during development, homeostasis, and disease [11,12,13,14]. Under steady-state conditions, BM-derived blood Ly6C^high^ monocytes give rise to two developmental streams: one with CCR2 expression, the acquisition of MHCII molecules, and a reduction in Ly6C expression; and the other with tissue-resident macrophages that become Ly6C^−^, CCR2^−^, and CX3CR1^+^, with a mixed expression of MHCII molecules [15].

Stress conditions within a tissue result in local proliferation of macrophage populations. However, there is a significant contribution from monocytes to the infiltrating myeloid cell pool during inflammation, e.g., a pathogenic infection or a growing tumor. Evidence from mouse models, such as breast [13], colorectal [16], and pancreatic cancer [17], has shown that macrophages in the TME have a mixed embryonic and BM origin, with differences in lineage markers, proliferation rates, and a dependence on growth factors, such as CSF1, as previously seen during development [10]. These findings argue that both monocyte infiltration and proliferation have roles in macrophage maintenance during tumor growth [18]. Nonetheless, such studies have not yet been thoroughly performed in melanoma, despite the potential impact they may have on directing therapeutic interventions [19,20].

Another critical factor influencing TAM function is their metabolic state, which impacts immune performance [21]. Glycolysis and mitochondrial oxidative phosphorylation (OXPHOS) are the two principal bioenergetic pathways in a cell, which directly or indirectly impact all other aspects of cellular metabolism. The cell-selective partitioning of nutrients and the metabolic processes in tumor-infiltrating immune cells could be exploited to design novel targeted therapies [22]. Both tumor and tumor-infiltrating immune cells rely on glucose, but it has been shown that cell-intrinsic programs drive the preferential acquisition of glucose and glutamine by immune and cancer cells, respectively [23]. A recent study [24] showed that a high cellular density in solid tumors can result in lactate build-up in its core. Lactate acts as a quorum-sensing-like signal, leading to increased oscillations in tumor cell HIF-1α activity, rescuing the inhibition of cell division and changing gene expression. Our prior findings showed that lactate-driven, HIF-1α- dependent Arginase (Arg)-1 expression in TAMs has an important role in tumor growth [25], possibly via the production of polyamines, with a critical role in tumor cell proliferation.

All this evidence highlights the need for an in-depth understanding of the origin, evolution, and processes that sustain and promote tumor-supportive macrophages. These insights could provide a roadmap for alternative therapeutic interventions in melanoma [26], but currently there is limited data available. In the present study, we analyzed TAM origin and its relationship with function and tumor progression in the Yale University Mouse Melanoma (YUMM)1.7 model. We focused on profiling multiple factors contributing to macrophage heterogeneity in this fast-growing, aggressive melanoma model with clinically relevant genetic features, low T-cell infiltration, and a dynamic change in the myeloid infiltrate. We found a strong BM-dependency of melanoma TAMs, but interestingly, these BM-derived TAMs matched macrophage subsets originated from mixed ontogeny, which have been observed in other models. These results highlight the importance of environmental cues in determining functional TAM subsets and may suggest a functional convergence of tumor-supporting macrophages, regardless of their origin, across tumor types.

## 2. Materials and Methods

### 2.1. Animals

C57BL/6J (B6), B6.129P2-Myb^tm1^Cgn/TbndJ (Myb^fl/fl^), and B6.Cg-Tg^(Mx1-cre)1^Cgn/J (Mx-1^cre^) mice were purchased from Jackson Laboratories. C57BL/6-Tg^(UBC-GFP)30^Scha/J (B6.UBQ-GFP) mice were kindly provided by the Girardi Lab at Yale. Myb^fl/fl^Mx-1^cre/wt^ mice were generated by crossing parental strains. Genotyping was performed according to the Jax protocols for those strains. The mice were kept according to the standard housing conditions of the Yale Animal Resources Center in specific pathogen-free conditions, and were 8–10 weeks of age at the moment of starting the experiments. All the experiments were performed according to the approved protocols of the Yale University Institutional Animal Care and Use Committee (IACUC).

### 2.2. Cell Culture

Yale University Mouse Melanoma (YUMM)1.7 were kindly provided by the Bosenberg Lab at Yale. YUMM1.7 is a clinically relevant model since it carries the main human melanoma driver mutations, BRAFV600E, Pten^−/−^, and Cdkn2^−/−^ [27]. The PDVC57 cell line was kindly provided by the Dr. Llanos-Casanova, CIEMAT, Spain. The cell lines were grown in DMEM/F12 media supplemented with 10% FBS, 1% NEAA, 2 mM L-glutamine, 1% Pen/Strep, and 1% sodium pyruvate (Gibco, Life Technologies, UK).

### 2.3. Tumor Studies and Sample Processing

The mice were anesthetized with isoflurane and intradermally injected with 0.5 × 10^6^ YUMM1.7 cells in both flanks, and monitored according to the IACUC approved mouse protocol. For the experiments with PDVC57 tumors, 1.0 × 106 tumor cells were injected into both flanks. The tumor volume was calculated as 0.52 × length × width^2^. At the experimental endpoint, the mice were euthanized in a CO_2_ chamber and the tumors were resected and weighted before being processed. Briefly, the tumors were first cut and then chopped with a razor into 1 mm^3^ pieces. They were incubated in digestion buffer (1X PBS Ca^+^Mg^+^ containing 0.1 mg/mL of DNase I, Roche 05401127001, and 0.82 mg/mL of Collagenase IV (C. histolyticum C1889, Sigma-Aldrich, MO, USA) at 37 °C in an orbital shaker for 30 min. The samples were vortexed, kept on ice, and filtered in 40 µm cell strainer. The cells were resuspended in ACK lysis buffer at RT for 5 min. The samples were finally washed, resuspended, and counted for downstream processing.

### 2.4. Histology and Immunofluorescence

For hematoxylin/eosin (H&E) staining, the tumor samples were fixed in formaldehyde and paraffin-embedded at Yale Pathology core, according to standard protocols.

For immunofluorescence, freshly collected tumor samples were fixed in 2% paraformaldehyde ON, washed, transferred to 30% sucrose in PBS for 24 h, and finally placed in a plastic mold in optimal cutting temperature compound (OCT) at −80 °C. Ten μm sections, cut in a Leica cryostat, were washed in PBS and stained with 1:1000 Rabbit anti-GFP (ab6556, Abcam, MA, USA) and 1:75 Rat anti-mouse F4/80 AF647 (123122, BioLegend, CA, USA) overnight at 4 °C. Next, the samples were washed and stained with 1:400 Goat anti-Rabbit AF488, counterstained with DAPI, and analyzed in a Leica SP5 confocal microscope at Yale CCMI Imaging Core.

### 2.5. Flow Cytometry

We used this technique to characterize the YUMM1.7 immune and TAM subpopulations, as well as to fate-map the TAMs infiltrating both the YUMM1.7 and PDVC57 tumors. We used the following antibodies and dyes (clone, catalog#): CD45 PerCP (30-F11, 103132), CD45.1 APC-Cy7 (A20, 110716), CD45.2 APC (104, 109814), B220 FITC (RA3-6B2, 103206), CD3e FITC (KT3.1.1, 155604), NK1.1 FITC (PK136, 108706), CD11b BV421 (M1/70, 101236), CD11b PECy7 (M1/70, 101216), F4/80 PerCP (BM8, 123126) and F4/80 AF700 (BM8, 123130), CD11c PECy7 (N418, 117318), CD207 PE (4C7, 144204), CSF1R BV605 (AFS98, 135517), Ly6C APC (HK1.4, 128016), Ly6C Pacific Blue (HK1.4, 128014), Ly6G AF647 (1A8, 127610), Ly6G FITC (1A8, 127606), CX3CR1 BV605 (SA011F11, 149027), Arg1 APC (A1exF5, 12-3697-82), CD40 PE (3/23, 124609), CD86 PE-Dazzle594 (GL1,105042), CD64 PE-Dazzle594 (X54-5/7.1, 139320), CD24 BV605 (M1/69, 101827), CD103 BV711 (2E7, 121435), MHCII APC-Cy7 (M5/114.15.2, 107628), CD206 APC (C068C2, 141708) from BioLegend, Live/Dead eFluor506 (65-0866-18), iNOS AF488 (CXNFT, 53-5920- 82), iNOS APC (CXNFT, 17-5920-82) and iNOS PE (CXNFT, 12-5920-82) from Thermo Fisher, CCR2 PE (475301, FAB5538P-100), and Arg1 APC (658922, IC5868A) from R&D Systems. TAM single-cell suspensions, obtained as previously described, were stained in FACS Buffer (PBS 2% FBS). Briefly, the cells were incubated with FcBlock 1:200 (anti-CD16/CD32, eBiosciences, CA, USA), washed, and stained for extracellular markers. CytoFix/CytoPerm and Perm/Wash Buffer kits (BD, 554714) were used for the intracellular staining steps, according to the manufacturer’s instructions, and stained for intracellular antigens. The samples were analyzed on a LSRFortessa (BD Biosciences, CA, USA). Gating for the analysis was performed as described in Appendix A. Flow cytometry data were analyzed using FlowJo software (TreeStar Inc., OR, USA). The TAM phenotypes were further analyzed via a principal component analysis (PCA) using SIMCA16 (Sartorius, Sweden) and GraphPad Prism.

### 2.6. Bone Marrow (BM) Transplant

To study the melanoma TAM origin, we used two strategies to track BM-derived cells and identify them through the expression of GFP and/or CD45.2^+^. The Myb^fl/fl^Mx-1^cre/wt^ (CD45.1^+^) mice were administered 10 μg/g of Poly(I:C) (P1530, Sigma-Aldrich, MA, USA) i.p. every other day, for a total of 7 times, to induce the genetic depletion of BM precursors. The B6.UBQ-GFP (CD45.2^+^) mice were treated with 200 μg/g of Busulfan (cat#14843, Cayman Chemical, MI, USA) i.p. every day, for a total of 5 times, to induce the chemical ablation of BM precursors. For the adoptive transfer, 10 × 10^6^ BM cells, extracted from the femur and tibia of the B6.UBQ-GFP and B6 donor mice as previously described [28], were transferred retro-orbitally to the Myb^fl/fl^Mx-1^cre/wt^ and B6.UBQ-GFP mice, respectively. Blood samples were extracted from the transplanted mice to evaluate their chimerism (Myb^fl/fl^Mx-1^cre/wt^ every 4 weeks, B6.UBQ-GFP every week). YUMM1.7 tumors were injected after the transplant was stable (Myb^fl/fl^Mx-1^cre/wt^ after 12 weeks, B6.UBQ-GFP after 8 weeks) and the tumors were processed after 2 weeks, along with blood and skin samples. B6.UBQ-GFP-transplanted Myb^fl/fl^Mx-1^cre/wt^ mice were also used to assess the PDVC57 TAM infiltration.

### 2.7. Bulk RNA Sequencing (RNA-Seq) Analysis

To perform a detailed characterization of and comparison analysis between the F4/80^+^ TAM subpopulations, we analyzed their transcriptomes. For this, we sorted the YUMM1.7 TAMs from day 21 tumors, prepared as described above, without fixing them. This timepoint allowed us to sort the necessary cell numbers of each TAM subpopulation for sequencing. The TAMs were sorted in a FACSAria instrument (BD Biosciences) from singlets and Live^+^CD45^+^Lin^−^CD11b^+^ cells and F4/80^high^ and F4/80^low^ were collected and fixed for bulk RNA-seq. The samples were sent to the Yale Center for Genome Analysis (YCGA) for RNA extraction and sequencing. The bulk RNA-seq data were processed with YCGA’s standard alignment pipeline. Protein coding genes with at least ten counts across the samples were retained for downstream analyses. The DESeq2 package in R [29] was then used to correct for batch effects, calculate the differentially expressed genes (DEGs; adjusted *p*-value < 0.05 and |log_2_FC| > 1) across the experimental conditions, and embed the samples in a principal component analysis (PCA) space. The results were visualized using the ggplot2 package in R [30]. A gene set enrichment analysis on the DEGs was performed using the Generally Applicable Gene-set Enrichment (GAGE) package in R [31]. Gene Ontology (GO) biological processes and the Kyoto Encyclopedia of Genes and Genomes (KEGG) were used as reference databases. We performed the gene-set enrichment analysis (GSEA) with the online software from UCSD/Broad Institute [32].

### 2.8. Fluorescent Bead-Based Multiplex Protein Secretion Profiling

To analyze the functional profiles of the YUMM1.7 macrophages, namely to validate the findings from the transcriptomic data, we characterized the signals secreted by the TAM F4/80 subsets. For the supernatant collection, the sorted macrophages were plated at a density of 1 × 10^6^/mL in Petri dishes, incubated at 37 °C ON, and collected, centrifuged, and kept at −80 °C or directly submitted to Eve Technologies Corp (Calgary, AB, Canada). The samples were analyzed with the Mouse Cytokine/Chemokine 44-Plex Discovery Assay^®^ Array (MD44). Most of the detected analytes were within the dynamic range of the standard curves of each analyte, observing no saturation in the samples analyzed. For data analysis purposes, those presenting an out of range (OOR) measurement below the parameter logistic standard curve were systematically replaced with the lowest value obtained for a particular analyte, as per the suggestion of the company. The data were natural log-transformed to aid visualization. The samples were visualized and hierarchically clustered using the clustermap function from the Seaborn module in Python. The non-normalized data were embedded in two dimensions using a PCA, as implemented in the multivariate statsmodels module in Python.

### 2.9. Statistical Analysis

For group comparisons, one-way or two-way ANOVAs with post hoc comparisons and linear regressions were performed in GraphPad Prism 9. Significance annotation: ns = not significant, * *p* < 0.05, ** *p* < 0.01, *** *p* < 0.001, and **** *p* < 0.0001.

## 3. Results

### 3.1. F4/80 Expression Defined YUMM1.7 Melanoma TAM Subsets with Partial Similarities with Skin-Resident Macrophages That Evolved during Tumor Progression

In the C57Bl/6J mice, the YUMM1.7 cells generated fast-growing i.d. tumors, which infiltrated the whole skin compartment, colonizing the dermis and displaying an inflamed epidermis (Appendix A). We performed an analysis of the myeloid infiltrate in the YUMM1.7 tumors. In the 14-day i.d. tumors, we detected an overall low immune infiltration, with a majority of myeloid cells, which was consistent with what had been previously described [33]. F4/80^+^ cells were the predominant myeloid subset (Figure 1a). When looking at Lin^−^MHCII^+^ cells, only a small fraction of them corresponded to dendritic cells (cDC1 and cDC2) or Langerhans cells, confirming TAM prevalence (Appendix A). CD11b^+^F4/80^+^ macrophages were found in both the YUMM1.7 tumors and normal skin (Figure 1b). Looking closer into the F4/80^+^ TAMs, we could determine F4/80^low^- and F4/80^high^-expressing subsets, as previously described in the skin and other compartments [10]. F4/80^high^ TAMs represented a smaller fraction of the total compared to F4/80^low^ and F4/80^neg^ TAMs (Figure 1c,d). The macrophage F4/80-based subset distribution was mirrored in the skin. Although TAMs were found throughout the tumors, either being evenly distributed or forming F4/80^+^ clusters, macrophages were only sparse in the skin compartment (Appendix A).

To validate the F4/80 subset annotation, we mapped the YUMMMER1.7 TAMs into the monocyte classification based on the Ly6C and MHCII expressions, known as the “monocyte waterfall” [34,35]. The F4/80^high^ subpopulation resembled skin tissue-resident macrophages, with no Ly6C expression; on the contrary, the F4/80^low^ subset aligned with monocyte-derived cells, predominantly Ly6C^+^MHCII^+^ (Figure 1e). We had previously determined that YUMM melanoma TAMs share features with both canonical in vitro M1-like and M2-like profiles, and that BMDMs can be shaped into a TAM-like state through cell–cell communications and interactions in a 3D environment [8]. To study melanoma TAMs in more detail, we phenotypically profiled macrophages from the tumors and normal skin using flow cytometry (Appendix A). We used principal component analysis (PCA) to compare these multi-dimensional phenotypic profiles. PC1 separated samples by F4/80 expression level, which explained most of the variance in these data, and clearly partitioned the TAM subsets (Figure 1f, Appendix A). F4/80^high^ TAMs were the most heterogeneous between the samples and more similar to the matched F4/80 skin subsets (Figure 1f).

To understand the origin of the differences in the F4/80 TAMs in the i.d.-injected tumors, we examined the individual marker expressions between the tumor- vs. skin-infiltrating macrophages. Skin-resident macrophage chemokine receptor expression patterns showed F4/80 expression to be inversely correlated with CCR2 expression, but positively associated with the expression of CX3CR1 (Figure 1g). Although the TAM subsets had differences in their chemokine receptor expressions, they did not follow the same pattern as the skin-resident cells, with an overall higher co-expression of both markers (Figure 1g). Similarly, the skin monocyte-derived cells had decreasing levels of CSF1R expression as they became more “mature” cells (i.e., MHCII^+^ F4/80^high^), while the TAMs retained significant levels of CSF1R expression and there were less MHCII^−^ cells as the F4/80 expression increased (Appendix A).

To characterize the dynamics of the F4/80 TAM subsets in the melanoma tumors, we then looked into how these subpopulations changed over time. Interestingly, the F4/80^+^ TAM fraction was enriched from day 7 to day 21, primarily due to a significant increase in the F4/80^high^ subpopulation (Figure 1h,i). The TAM subsets also exhibited a divergence in their phenotype as the tumor progressed (Figure 1j, Appendix A). When we looked into the canonical M1/M2 markers, there was no clear association between our TAM subsets, or their evolution, with canonical macrophage polarization states, as we have previously described for the general YUMM1.7 TAM population [8]. Higher expressions of iNOS and MHCII were present in more immunosuppressive subsets and at advanced timepoints (Figure 1k,l). Interestingly, a higher percentage of iNOS expression did not necessarily correlate with a higher mean fluorescent intensity, potentially impacting the overall immune performance (Appendix A).

### 3.2. Despite Exhibiting Some Skin-Resident-like Features, Melanoma TAMs Originated Almost Exclusively from Circulating Monocytes

Given the partial similarities between melanoma TAMs and skin-resident cells, in addition to the observations made in other tumor types [13,16,17,36], we further investigated the TAM ontogeny to better understand their immune function. For this, we used two strategies to track BM precursors and tumor infiltration (Figure 2a and Appendix A). Chemical BM ablation induced by busulfan generated a model with an almost complete and homogenous BM transplant, while the genetic depletion of the Myb^+^ BM precursor generated a graded engraftment success, allowing for an evaluation of the potential competition between host and donor cells (Figure 2b and Appendix A). The busulfan-treated mice showed significantly better myeloid cell engraftment, observed in the circulation, which allowed us to evaluate the F4/80 TAM origin.

Using the first approach through chemical ablation, we could determine that, by day 14, all the YUMM1.7 TAMs originated from GFP^−^ donor BM precursors, by comparing the cells in the circulation with the tumor-infiltrating ones (Figure 2c and Appendix A). We then studied the TAM infiltration in detail using the genetic ablation model, Myb^fl/fl^Mx-1^cre^. There were no differences in the YUMM1.7 tumor growth between the BM-engrafted and non-engrafted mice, and all F4/80 subsets were observed in the infiltrate (Appendix A). We once more used the chimerism observed in the circulating cells to analyze the tumor and skin myeloid infiltration (Appendix A). The evidence showed that all the melanoma TAMs were derived from BM precursors; the chimerism of each TAM subset matched the host/donor monocyte proportion in circulation (Figure 2d), in the wide spectrum of transplant engraftment. Using this same model, we validated the macrophage infiltration in the mouse skin (Appendix A). The F4/80^high^ macrophages, previously described to be Myb-independent during development and to colonize the tissue and self-renew [10], showed lower levels of chimerism in the skin compared to monocytes in circulation (Appendix A). We also analyzed the TAMs in a non-melanoma skin cancer model, PDVC57 [37], to compare the macrophage infiltration in the dermal compartment (Appendix A). In the PDVC57 tumors, we observed the same F4/80-based TAM subsets, as in YUMM1.7 tumors. But, interestingly, while the F4/80^low^ TAMs exclusively originated from BM precursors, the F4/80^high^ TAMs showed a mixed origin. Based on these observations in the Myb^fl/fl^Mx-1^cre^ model, the TAM origin in YUMM1.7 could not be exclusively attributed to the dermal location of the tumors. We further validated the macrophage infiltration via tissue immunofluorescence. Shown in Figure 2e, increasing levels of chimerism in the Myb^fl/fl^Mx-1^cre/wt^ mice corresponded with a higher amount of GFP^+^ F4/80^+^ in the tumor, while the Mx-1^wt/wt^ controls, with minimal engraftment, showed no F4/80^+^ donor cells. In the busulfan-treated mice, with over 95% of myeloid donor cells, only non-immune stromal cells were GFP^+^ (Appendix A).

### 3.3. Melanoma F4/80^+^ TAM Subsets Have Distinct Immunosuppressive Profiles with Specific Active Metabolic and Functional Pathways

Despite having a common BM origin, the F4/80^+^ melanoma TAMs evolved into phenotypically diverging subsets. To understand in more detail the differences between these F4/80^+^ TAM subsets, and their relationship with immunosuppression and tumor-supportive functions, we performed bulk RNA-seq. We collected data from the F4/80^high^ and F4/80^low^ TAM subsets sorted from the YUMM1.7 tumors in three independent experiments, with approximately 0.15–1.50 × 10^6^ cells per replicate (Appendix A). Considering only protein-coding genes, the F4/80^high^ and F4/80^low^ samples were separated in a two-dimensional space by PCA and unsupervised hierarchical clustering revealed distinct transcriptional patterns across the TAM subsets (Figure 3a and Appendix A). The identification of the differentially expressed genes (DEGs) revealed key transcriptional differences between the F4/80^high^ and F4/80^low^ melanoma TAMs (Appendix A, top 50). Interestingly, both TAM subsets expressed mutually exclusive groups of M2-like immunosuppressive markers. Of note was that the F4/80^low^ TAMs overexpressed *Chil3*, *Mmp9*, and *Ear2*, while the F4/80^high^ TAMs had elevated expressions of the scavenger receptors *Mrc1*, *Mertk*, and *Cd163* (Figure 3b). Each TAM subset also showed an upregulation of specific chemokines and chemokine receptors. Most importantly, these two TAM subpopulations have gene profiles similar to TAM clusters previously identified in other tumor models and the human setting [26,38,39,40].

We further identified differences between the cellular and metabolic programs of the F4/80^high^ and F4/80^low^ TAM subsets. Gene-set enrichment analysis (GSEA) (Appendix A) revealed that the F4/80^high^ TAMs had upregulated oxidative phosphorylation, with increased expression of *Igf1* and *Cd38*, and lipid metabolism through the peroxisome pathway, while also showing increased proliferation pathways, such as the G2M checkpoint, as well as protein secretion (Figure 3b,c,e and Appendix A). In contrast, the F4/80^low^ cells were enriched for proinflammatory signaling pathways, such as TNFα signaling via NFκB, IFNα, and IFNγ signaling, along with responses to hypoxia and angiogenesis. These also showed expression of glycolysis-associated ABC- and glutamate-transporter genes, along with upregulated glycolysis KEGG metabolic pathways (Figure 3d,e and Appendix A). These results highlighted important differences in terms of the metabolic heterogeneity, proliferative capacity, and self-renewal of the YUMM1.7 melanoma TAMs.

To validate the functional differences and pathways inferred from the RNA-seq analysis, we screened for the secretion of cytokines and chemokines with relevant immune functions in the TME. For this purpose, we profiled the supernatants from sorted F4/80^high^, F4/80^low^, and F4/80^neg^ melanoma TAM subsets, through a multiplex bead-based secretion assay. Protein secretion showed differences in the melanoma F4/80 TAM subsets for growth factors, pro-inflammatory and immunosuppressive cytokines, and immune-trafficking chemokines. PCA-embedding of the secretion data showed that, despite the heterogeneity across the replicates, the F4/80 TAM subsets were separated along PC2 (Figure 3f). The PC coefficients identified factors contributing to the differences across the TAM subsets (Appendix A). To identify the potential drivers of the functional divergence between the F4/80 subsets, we looked into the secreted factors individually (Appendix A). When contrasting the F4/80^high^ and F4/80^low^ TAMs, F4/80^high^ showed significantly higher levels of IL-15, IL-16, and CX3CL1, and higher trends for IFNβ1, IL-1α, CCL3, CXCL5, IL-20, M-CSF, and IL-4, while F4/80^low^ secreted more VEGF, GM-CSF, IL-1β, IL-10, CCL2, CXCL1, CXCL2, CXCL10, and CCL22, with trends for G-CSF, TNF-α, IFN-γ, IL-5, IL-9, and CCL11 (Appendix A). These profiles partially validated some of the transcriptome profiles, adding to the phenotypic differences established using flow cytometry. Finally, we hierarchically clustered the bulk secretion data to define patterns in the trends contributing to the differences in the F4/80 TAM subsets. We identified seven functional clusters, determining an overall low or high secretion across the subsets (clusters one and four) and a clear trend across the subsets (clusters two and five), while the others were less defined and harder to interpret, primarily due to sample-to-sample heterogeneity (Figure 3g).

## 4. Discussion

In the present study, we analyzed the YUMM1.7 melanoma model and identified distinct TAM subsets based on their F4/80 expressions. Interestingly, they partially matched prior monocyte/macrophage classification systems, but there was a need to better understand the melanoma TAM origin and dynamics during tumor progression. The F4/80^high^ TAM fraction was shown to increase over time and a display tissue-resident-like phenotype. Previous seminal work [10] has shown that F4/80^bright^ and yolk-sac macrophages have increased expressions of receptors *Cx3cr1* and *Csf1*, along with proliferation markers. However, the key phenotypic markers in YUMM1.7 F4/80^high^ TAMs, such as CSF1R and chemokine receptors, showed strong differences, which make them resemble recruited monocytes. The RNA-seq signatures we identified in the melanoma F4/80 subsets could be mapped into TAM clusters that were recently identified through scRNAseq gene signatures from different tumor types and pathologies, most of which were shown to have mixed ontogeny [26,38,39,41,42]. Interestingly, the fate-mapping of the monocytes determined that all the melanoma TAMs had a BM origin, although they differentiated over time into two separate subsets with different phenotypic profiles. Furthermore, the transcriptional profiling showed the F4/80^high^ subset to have features resembling tissue-resident macrophages from an embryonic origin, such as microglia, a phenomenon that has been seen in other tissue contexts [43]. These results emphasize the role of environmental cues in shaping macrophage identity and function, with BM-derived melanoma TAMs rapidly mimicking skin-resident cells.

More importantly, our results may suggest a functional convergence of tumor-supporting macrophages, regardless of their origin, across tumor types. Several groups have studied macrophage ontogeny in other tumor types, but mostly in orthotopic settings [13,16,17,40]. Here, in the dermal compartment, we observed a different TAM ontogeny and composition based on the skin tumor type analyzed. In order to fully understand how F4/80^+^ TAM subsets and evolution are driven and determined by the skin microenvironment and the contribution of encoded factors within each specific TME, further studies are required. However, recent spatial mapping of the immune landscape in tumors has shown that completely different functional immune environments can be a few millimeters away [44], and that the specific location in the TME and the location of the tumor itself influence immune phenotype and function [36,45,46]. We observed differences in the TAM distribution within different regions of the YUMM1.7 TME. A thorough mapping of melanoma TAMs, through the markers we and others have identified in each F4/80 subset within the immune neighborhoods and functional microdomains in the TME, could help understand their cell–cell interactions and evolution during tumor progression, and assess novel targeted therapies in order to improve future treatment rationale [19,47,48,49].

In addition to the contributions provided in this study on melanoma TAM origin and characterization, there are a few interesting aspects that may help to direct novel therapeutic interventions. A prior study from our group with a genetically engineered mouse model (GEMM), which, in turn, originated the YUMM cell lines, showed differences in the F4/80 expression in TAMs, which secreted Chi3l3, MMP9, and IGF1, among other factors [4], which were modulated by administering CD40-agonist and contributed to tumor control. In this line, the functional divergence of these F4/80^+^ TAMs, with distinct active metabolic pathways, opens up another road for discovering therapeutic approaches [21,23,50]. Tissue-resident-like TAMs, which have a longer lifespan and the potential to self-renew and expand through proliferation, could be an ideal candidate to target in the TME, but may be more difficult to be externally reprogramed into tumoricidal cells with traditional immunotherapies. Metabolic state and energy sources have been shown to shape macrophage function in the TME [51,52]. With the logical limitations, a metabolic reprogramming approach gains more importance in this context, since it could target cell-intrinsic programs that could impact directly on immune function [53,54,55]. New approaches, making use of metabolic redundancy, have been tested in inflammatory settings [56]. The skin compartment may provide an ideal setting for testing novel approaches through topical application [57], since it may prevent an otherwise high systemic toxicity.

## 5. Conclusions

The results presented in this report provide an in-depth characterization of the evolution of melanoma TAMs in a mouse model that resembles the human disease. Unlike other previously studied orthotopic tumor models, such as lung or pancreatic cancer, i.d. melanomas exhibited primarily BM-derived TAMs. More interestingly, they presented in discrete, phenotypically and functionally distinct subsets, with evolving ratios. We comprehensively analyzed the melanoma TAM ontogeny and how the discrete TAM subsets, defined by their F4/80 expressions, showed distinct cellular and functional profiles from their early stages and continued to evolve throughout tumor progression.

The TAM similarities between tumor models might help to make inferences about the TAM evolution in the TME and their heterogeneity at the single-cell level. More importantly, we could speculate about potential treatment options. A major challenge as we look forward, though, is centered on how to translate these investigations on macrophage origins, differentiation, and maintenance into humans. Fortunately, in recent years, there have been advances in several tumor types trying to validate the discoveries made with mouse models [26,36,40,45]. Ultimately, we want to have a better understanding of the heterogeneity and functional characteristics of human TAMs to improve prognoses and treatment options.

## Figures and Tables

**Figure 1 cancers-15-03330-f001:**
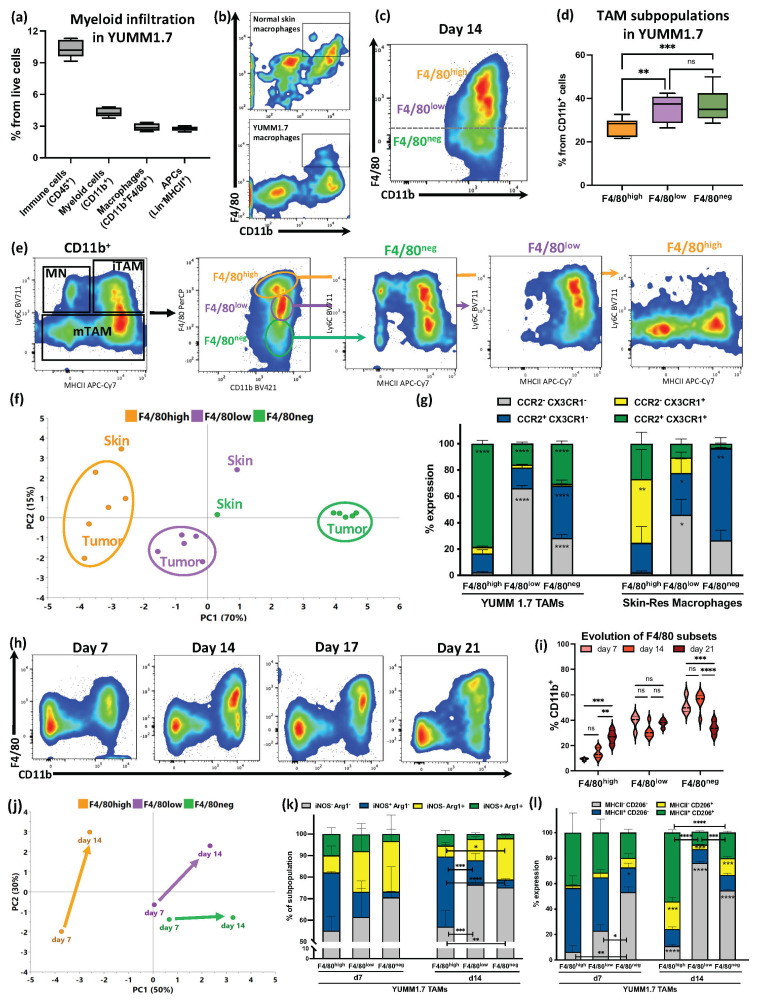
Characterization and evolution of F4/80^+^ tumor-associated macrophage subsets in intradermal melanoma tumors. (**a**) Box plots of YUMM1.7 immune infiltration, quantified by flow cytometry. Percentage of total immune infiltration (CD45^+^), myeloid cells (CD45^+^CD11b^+^), and APC (CD45^+^Lin^−^MHCII^+^). n = 10, pooled from at least 3 independent experiments. (**b**) Representative flow cytometry plots gating the macrophages (CD11b^+^F4/80^+^) out of the total CD45^+^ cells, both in normal skin and in YUMM1.7 i.d. tumors. (**c**) Representative plot of the Lin^−^CD11b^+^ myeloid subpopulations determined by their F4/80 expression in YUMM1.7 melanoma tumors. (**d**) Box plots showing F4/80^+/−^ myeloid subpopulations at day 14 YUMM1.7 i.d. tumors. n = 15, pooled from 3 independent experiments. (**e**) Representative flow cytometry plots of the expression of Ly6C and MHCII markers (‘monocyte waterfall’), defining myeloid cell subsets, broken down into the different F4/80^+/−^ subpopulations in YUMM1.7 tumors. MN = monocytes; iTAM = immature TAMs; and mTAMs = mature TAMs. (**f**) PCA plot of the tumor-infiltrating and skin-resident F4/80 subpopulations phenotype, assessed using flow cytometry. Details of flow panels and parameters included in the PCA can be found in Appendix A. Five independent tumor samples from day 14 were included in the analysis. Skin macrophage phenotype data are plotted as reference subpopulations, average of 2 independent samples. (**g**) Expression of chemokine receptors CCR2/CX3CR1 in F4/80 macrophage subsets in YUMM1.7 tumors, compared to skin-resident macrophage subsets. Statistical analysis showing comparisons between marker-expressing subsets within each macrophage type. TAMs n = 5 and Skin n = 3, pooled from at least 2 independent experiments. (**h**) Representative flow cytometry plots showing the changes of TAM subpopulations over time in YUMM1.7 i.d. tumors. (**i**) Quantification of the evolution of the F4/80 TAM subpopulations, assessed using flow cytometry, at days 7, 14, and 21 after YUMM1.7 i.d. tumor injection. n = 4–6, pooled from independent experiments. (**j**) PCA plot of tumor-infiltrating F4/80 subpopulations phenotype at day 7 and day 14 YUMM1.7 i.d. tumors in B6 mice. Details of flow panels and parameters included in the PCA can be found in Appendix A. For day 7 tumors, n = 3 and day 14 tumors, n = 5. (**k**,**l**) Analysis of canonical M1/M2-marker pairs co-expression (iNOS/Arginase-1 and MHCII/CD206) in TAM subpopulations from YUMM1.7 tumors, at day 7 and 14. Statistical analysis showing comparisons between marker-expressing subsets within each timepoint. n = 3–5 pooled mice per group, from at least 2 independent experiments. Ns = not significant, * *p* < 0.05, ** *p* < 0.01, *** *p* < 0.001, and **** *p* < 0.0001.

**Figure 2 cancers-15-03330-f002:**
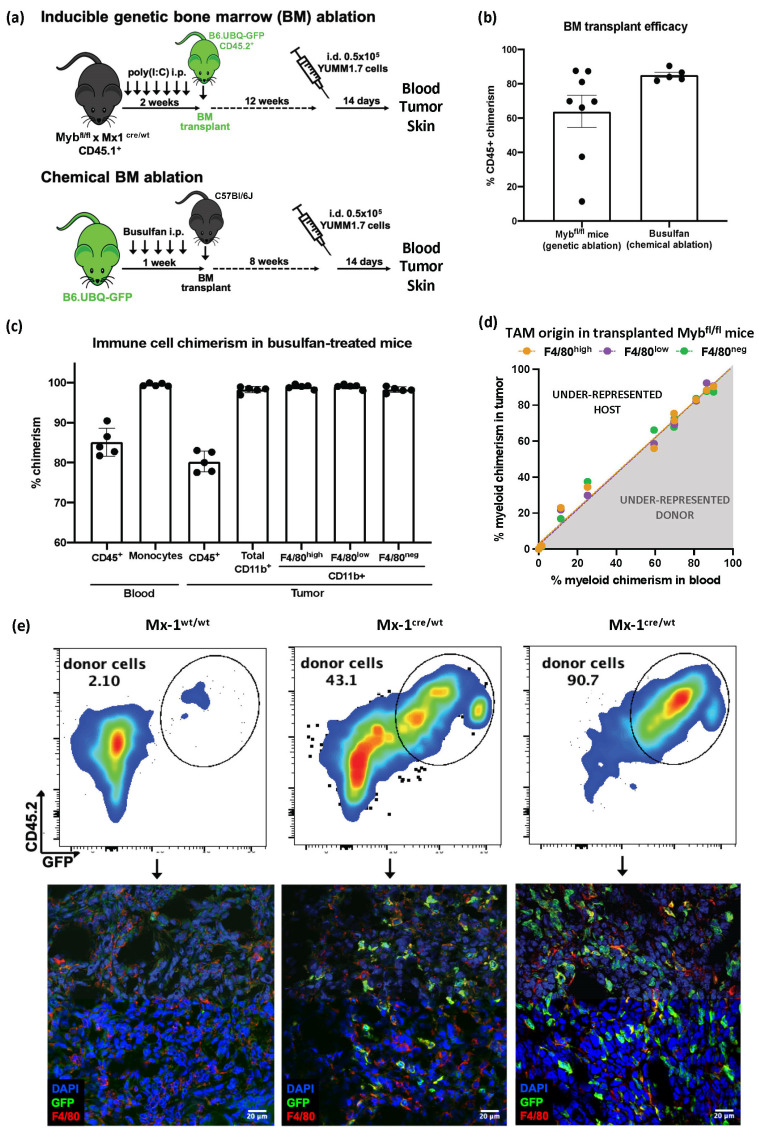
Origin of tumor-infiltrating myeloid cells in YUMM1.7 melanoma tumors. (**a**) Schematic design showing two different strategies used to perform bone-marrow (BM) transplants and track the origin of YUMM1.7-infiltrating macrophages. (**b**) BM transplant efficacy shown as percentage of chimerism of CD45^+^ cells in transplanted mice blood. Chimerism is expressed as the percentage of donor cells in the host, in both BM transplant approaches. (**c**) Tumor-infiltrating myeloid cell origin in YUMM1.7 tumors in busulfan-treated mice. Percentages of chimerism of total CD45^+^ cells and myeloid cells were compared between blood and tumor compartments. n = 5. (**d**) Assessment of macrophage origin in YUMM1.7 tumors in Myb^fl/fl^Mx-1^cre/wt^ and Myb^fl/fl^Mx-1^wt/wt^ mice. Linear regressions were performed to compare the chimerism of myeloid cells observed in circulation and the chimerism of myeloid tumor-infiltrating cells. TAM subsets were analyzed separately. Tumor samples, n = 12, pooled from 3 independent experiments. (**e**) Macrophage infiltration and chimerism were validated using immunofluorescence on corresponding tumor slides in Myb^fl/fl^ mice with increasing levels of GFP^+^ BM donor engraftment. Donor cells and TAMs were detected by GFP and F4/80 expression, respectively.

**Figure 3 cancers-15-03330-f003:**
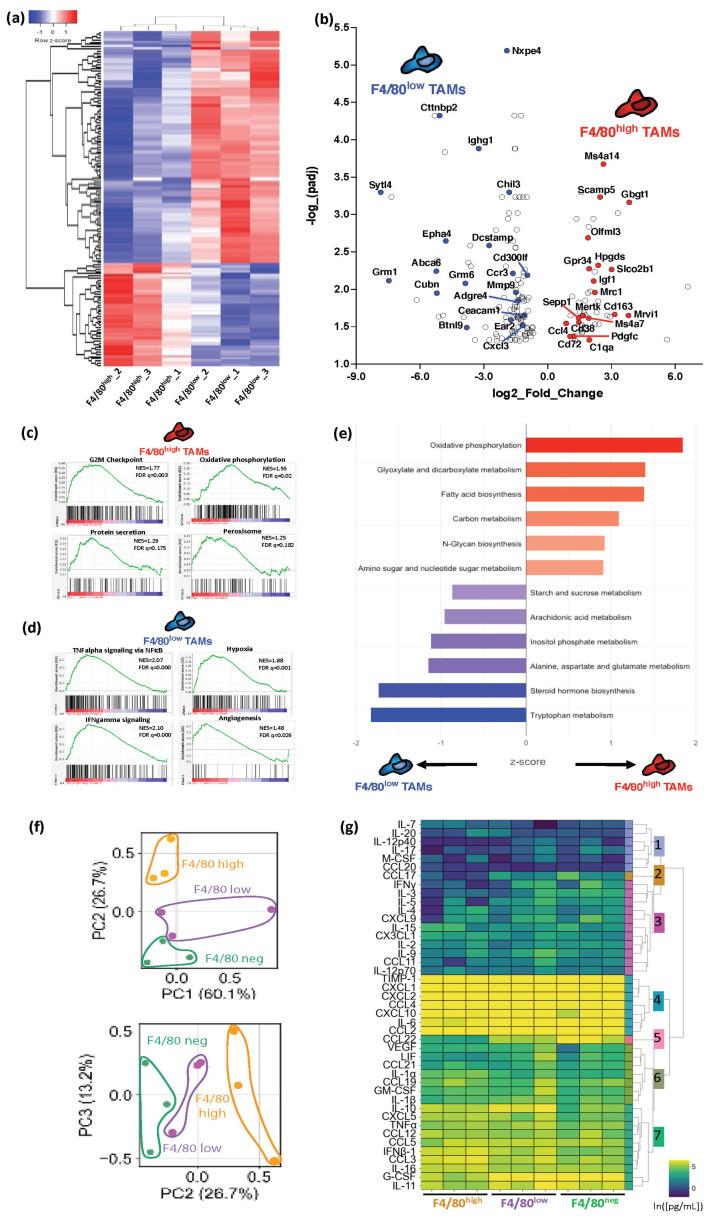
Functional characterization and profiling of TAM subsets in YUMM1.7 melanoma tumors. (**a**) Heatmap with hierarchical clustering of TAM subpopulations based on z-score-transformed gene expression from bulk RNA-seq. F4/80^high^ and F4/80^low^ subsets were isolated from 10 pooled tumors and RNA-seq data were collected in 3 independent experiments. (**b**) Volcano plot showing the top 50 differentially expressed genes (DEGs) between F4/80^high^ and F4/80^low^ subsets (*p*-adj < 0.05, abs(log2FoldChange(FC)) > 1). (**c**,**d**) GSEA Hallmark gene sets from the mouse MSigDB, with an FDR < 25% for the F4/80^high^ and F4/80^low^ TAM DEGs (NES = normalized enrichment score, FDR = false discovery rate). (**e**) Enrichment of F4/80^high^ (right, red) and F4/80^low^ (left, blue) DEGs (protein-coding genes only) for KEGG metabolic pathway terms. Bar lengths represent associated z-score. (**f**) PCA embedding of TAM samples from cytokine/chemokine functional profiling, assessed by bead-based multiplex secretion assay (PC2 vs. PC1, top; PC3 vs. PC2, bottom). Samples clearly separated by TAM subset along PC2, which explained 26.7% of variability in the data. (**g**) Heatmap of natural log-transformed protein secretion from (**f**). Hierarchical clustering identified 7 functional clusters across TAM subsets.

## Data Availability

The datasets generated and analyzed in this study are publicly available and can be found in the GEO database: https://www.ncbi.nlm.nih.gov/geo/query/acc.cgi?acc=GSE223545, accessed on 30 January 2023.

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
