# Peer review of "Functionally and Metabolically Divergent Melanoma-Associated Macrophages Originate from Common Bone-Marrow Precursors"

_cancers, 2023, doi:10.3390/cancers15133330_

Round 1

Reviewer 1 Report

The authors show an in-depth analysis of tumor-associated macrophages in a melanoma mouse model and cell line. The introduction is excellent in describing the relevant context, but it might help to add some more words on what knowledge is lacking and how this paper is going to answer that. 

I would advise a few clarifications to the methods. I had difficulties from the description of the methods to understand which technique was performed on which cell line or mouse material or from the cell lines injected in the mice. Maybe also some extra words on why you decided on designing this study with these techniques in the methods section. So that is it more clear for which question which experiment is used. 

For the results I would prefer not to put references here. For g, supplement I would also like to see the corresponding HE or CD68. What is the definition of F4/80 neg, low and high? In the text I can only read that this was clear from figure 1c-d. The separation of these groups in not so clear for me from those figures and many of the further analysis rely on this, so please clarify. 

In the discussion it seems that only a small part is highlighted and that it would help if it would give a summary of all messages from this paper and put it in a broader context. The discussion contains some far-fetched future directions in a role for immunotherapy. I think that a fundamental study like this does not need lines like that, it is relevant to understand more about tumor immunology and we see what that knowledge will bring in the future. But I know others might like these phrases, so I would only leave them out to add some more on the details of the study if there are word restrictions. 

Reviewer 2 Report

The Review “Functionally and metabolically divergent melanoma-associated macrophages originate from common bone-marrow precursors” by Pizzurro et al.

The Manuscript by Pizzurro et al. investigate the origin and differences of macrophage infiltration in normal and melanoma microenvironment. The Manuscript is well written and logically organized. The main text includes essential information describing the origin, gene expression, and phenotype of tissue-resident macrophages, and other molecular mechanisms of cancer progression. The Manuscript is addressing an important direction in immunosuppressive tumor microenvironment of cancer.

My minor suggestions to the authors are the following:

1.     The Results sections lacking a short description of goals of the preformed experiments.

2.     Discussion section it confusing – authors reviewing data from previous publications from them and others with little discussion of current results. All findings discussion is summarized in Conclusion – probably it would be beneficial for the manuscript to combine these sections.

3.     At several places authors speculate on impact of their data on future prognosis and treatment options:

“These findings provide evidence for potentially targeting specific immunosuppressive TAMs in advanced tumor stages.” “These findings provide a deep understanding of the evolution of melanoma-supportive macrophage subsets, with implications for targeted therapeutic strategies improve melanoma treatment.” “Ultimately, we want to have a better understanding of the heterogeneity and functional characteristics of human TAMs to improve prognosis and treatment options.”

Potentially, this statement is true, but the Manuscript lack information on proposed therapeutic approaches rising from these findings. I suggest toning down some sentences by removing the unsupported speculation or adding the missing information for at least the most remarkable points.

Round 2

Reviewer 1 Report

No more comments.